# Feedback Graph Attention Convolutional Network for MR Images Enhancement by Exploring Self-Similarity Features

| | |
|---|---|
| **Xiaobin Hu**[*1] | XIAOBIN.HU@TUM.DE |
| **Yanyang Yan**[*2] | YANYANYANG@IIE.AC.CN |
| **Wenqi Ren**[†2] | RWQ.RENWENQI@GMAIL.COM |
| **Hongwei Li**[1] | HONGWEI.LI@TUM.DE |
| **Amirhossein Bayat**[1] | AMIR.BAYAT@TUM.DE |
| **Yu Zhao**[†1] | YUZHAO90@OUTLOOK.COM |
| **Bjoern Menze**[1] | BJOERN.MENZE@TUM.DE |

[1] *Department of Computer Science, Technische Universität München, Munich*

[2] *Institute of Information Engineering, Chinese Academy of Sciences, Beijing, China*

## Abstract

Artifacts, blur, and noise are the common distortions degrading MRI images during the acquisition process, and deep neural networks have been demonstrated to help in improving image quality. To well exploit global structural information and self-similarity details, we propose a novel MR image enhancement network, named Feedback Graph Attention Convolutional Network (FB-GACN). As a key innovation, we consider the global structure of an image by building a graph network from image sub-regions that we consider to be node features, linking them non-locally according to their similarity. The proposed model consists of three main parts: 1) The parallel graph similarity branch and content branch, where the graph similarity branch aims at exploiting the similarity and symmetry across different image sub-regions in low-resolution feature space and provides additional priors for the content branch to enhance texture details. 2) A feedback mechanism with a recurrent structure to refine low-level representations with high-level information and generate powerful high-level texture details by handling the feedback connections. 3) A reconstruction to remove the artifacts and recover super-resolution images by using the estimated sub-region self-similarity priors obtained from the graph similarity branch. We evaluate our method on two image enhancement tasks: i) cross-protocol super resolution of diffusion MRI; ii) artifact removal of FLAIR MR images. Experimental results demonstrate that the proposed algorithm outperforms the state-of-the-art methods.

**Keywords:** Magnetic resonance imaging, image enhancement, self-similarity, graph similarity branch, feedback mechanism.

## 1. Introduction

For Magnetic Resonance Imaging (MRI) sequences, it is an inevitable dilemma to achieve a balance between image resolution, signal-to-noise ratio, and acquisition time (Brown et al., 2014). Higher resolution imaging grasps more structural details and provides more diagnostic information, but requires longer acquisition time (Sui et al., 2019). Since the signal-to-noise ratio is proportional to the slice thickness and the square root of scanning time, the

---

[*] Contributed equally

[†] indicates the corresponding authors

longer acquisition time leads to the performance drop of the signal-to-noise ratio and tends to generate artifacts caused by physiologic motion such as respiratory motion and physical movement of subjects. Considering the limited and costly MRI resource, some thick slices and low scan time MRI images are usually utilized to get a desired signal-to-noise ratio (Lee et al., 2020; Wu et al., 2019; Meurée et al., 2019). Consequently, the use of image enhancement techniques is an established field of research in medical image computing and imaging physics (Shi et al., 2015), for example, to prevent blurring and information loss when co-aligning different image volumes in a multi-parametric sequence.

Recently, Convolutional Neural Network (CNN) based approaches have shown dramatic improvements over traditional super-resolution (SR) methods and exhibited state-of-the-art performance in natural and medical images. A super-resolution convolutional neural network (SRCNN) (Dong et al., 2014) was proposed to learn a nonlinear mapping between the low-resolution (LR) and high-resolution (HR) images. Wide residual networks with fixed skip connections (Shi et al., 2018) was presented for MR images super-resolution. A new CNN-based model (Tanno et al., 2017) was proposed for a diffusion tensor imaging SR task. Besides, Graph Neural Networks (GNN) have also shown their powerful ability to exploit structural information dealing with data of graph structure. The notation of GNN was firstly introduced (Gori et al., 2005), and then further elaborated as a generalization of recursive neural networks, which is widely used to explore the structural characters in various applications including chemistry, recommender systems, and social network study to deal with challenge tasks, e.g., finding the chemical compounds that are most similar to a query compound, tackling the graph similarity computation for query systems (Bai et al., 2019). Nowadays, it is an interesting trend to combine GNN and CNN to develop their corresponding advantages (Veličković et al., 2018). GNNs help with reducing the data dimensionality from image features extracted by CNN to high-level and compact features in graph nodes. FCNs are limited in the receptive field. Adding a GNNs could increase the receptive field of networks when dealing with large images. The combination of CCN and GNN is a convolutional graph neural network that generalizes the operation of convolution from grid data to graph data. It plays a central role in building up many complex GNN models (Wu et al., 2020).

To avoid generating inconsistent HR results after replacing the LR patches, in our method, the similar patch pairs are matched in feature space and the graph attention mechanism is used to update features representation of each patch (node) with the adaptive weight combination of those similar patches' features. As far as we know, it is the first work to explore the self-similarity and continuous relationship of MRI and fully exploit the feedback mechanism to increase the reconstruction accuracy for MR images. More specifically, in this paper, we propose a novel biomedical image enhancement network based on the feedback mechanism and graph attention convolutional network, where graph networks are employed as a self-similarity strategy which assigns larger weights to the more important and similar nodes or features.

The main contributions of this paper are:

1) We propose a Feedback Graph Attention Convolutional Network (FB-GACN) for MR image enhancement. To the best of our knowledge, it is the first work to construct a graph-based network into the image enhancement by exploring globally structural similarity among similar paired sub-regions.

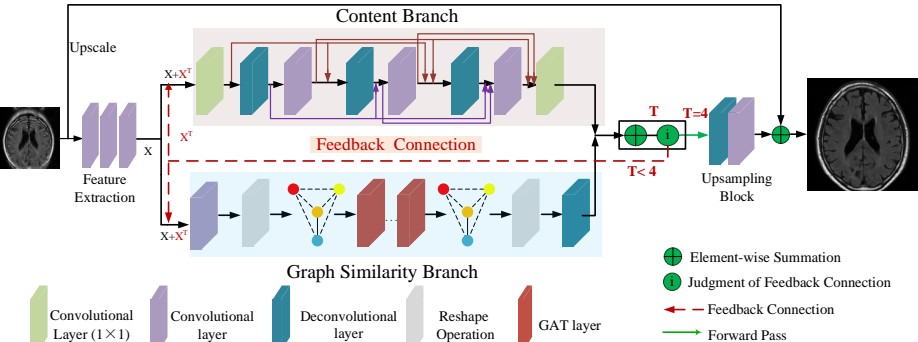

Figure 1: Architecture of the proposed FB-GACN model. Our FB-GACN contains three parts: 1) The content block to generate the high-level texture details. 2) The graph attention branch to exploit the similarity and symmetric knowledge across MRI patches. 3) A reconstruction to remove the artifact and reconstruct super-resolution MRI by using the estimated patch correlation priors. The feedback mechanism is the recurrent structure to refine $x$ features with high-level $x^T$ by the feedback connections.

2) We propose a self-similarity learning strategy to update the features of each node in a graph. Learning the symmetry and similarity relationship of each pair, the content with same texture (e.g., edges, corners, and lesions) gets sharper and can be used to remove some artifacts. It recovers more texture details by employing the feedback mechanism (consecutive iterations) to facilitate LR images to reconstruct SR images.

3) We demonstrate the performance in two crucial tasks: i) cross-protocol super resolution of diffusion MRI and ii) MRI artifacts removal. The proposed network achieves better high-resolution criteria and superior visual quality compared to state-of-the-art methods.

## 2. Method

The whole pipeline consists of following three steps. Firstly, a stack of convolution layers extracts the low-resolution features of input distortion images. Afterward, the content branch and graph similarity branch work parallel to exploit the texture and self-similarity information. Finally, the upsampling block reconstructs final super-resolution results using the estimated patch correlation and texture priors.

**Specialized design for MR images**: Our method aims to learn the symmetry and self-similarity relationship of patch-based features in multi-modal brain MR images where the structure of the brain is normally symmetry, shown in Fig. 2 (a). To meet this requirement, we designed a specialized Graph-based structure to merge the high-similarity information of sub-regions by updating larger weights to the more important and similar nodes or features in a graph attention fashion.

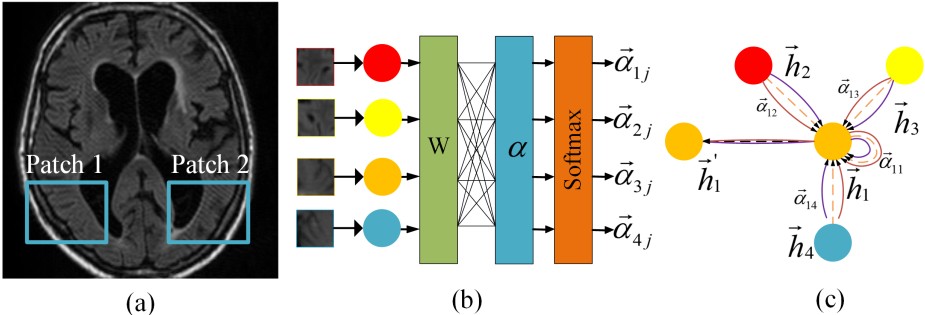

Figure 2: **(a)** Exploring the self-similarity features to remove artifacts: Swapping the artifacts features in Patch 2 with clear features of Patch 1. **(b)** The employed attention mechanism. A shared linear transformation $W$ is applied to every node. Afterwards, a self-attention mechanism $a$ is calculated on features to learn the correlation among nodes. **(c)** An illustration of multi-head attention mechanism by node 1 on its neighbors.

## 2.1. Architecture of FB-GACN

The structure of the proposed FB-GACN is illustrated in Fig. 1. A long skip connection is added to pass the upsampled LR image to the output result as we only want to learn the residual modifications. After feature extraction, the output are low-resolution features with the dimension of $h \times w \times d$, where $h$ and $w$ denote the spatial dimension of the LR input and $d$ is the number of feature channels. Then the LR features are imported into the content branch and graph similarity branch, respectively. The upsampling block $U$ is made up of deconvolution layers to upscale the HR features, and convolutional layers to recover a residual image. The final reconstruction SR images are the pixel-wise sum of the upsampled LR input and the residual image. The mathematical formulation is elaborated as:

$$I^{SR} = f_U \left[ f_G \left( f_E \left( I^{LR} \right) \right) + f_F \left( f_E \left( I^{LR} \right) \right) \right] + I_{up}^{LR}, \tag{1}$$

where $f_E(\cdot)$, $f_G(\cdot)$, $f_F(\cdot)$, and $f_U(\cdot)$ represent the operations of the feature extraction $E$, graph similarity branch $G$, content branch $F$ and upsamling $U$ blocks, respectively. The objective function is $L_1$ norm-based loss function. The network is trained by minimizing the objective function as following:

$$\ell_{(\theta)} = \frac{1}{n} \sum_{i=1}^{n} \left\| I_i^{SR} - I_i^{HR} \right\|_1, \tag{2}$$

whre $\theta$ and $n$ are the parameters of the network and the number of images pairs, respectively. $I_i^{SR}$ is the reconstruction of super-resolution MRI, and $I_i^{HR}$ is the corresponding ground truth.

## 2.2. Graph Similarity Branch

Graph similarity branch employs graph attention network layers (GAT) (Veličković et al., 2018) to make use of the contextual information among image patches to help recover structure and remove artifacts. After feeding the extracted LR feature maps to a convolutional layer with stride of $s$ and kernel size of $p$, we form a graph using the $n \times d$ matrix where

we assume there exist $n$ nodes with $d$-th dimensional features. Each node is connected with five neighboring nodes and the attention coefficient of each node is updated. The single graph attention layer is shown in Fig. 2. The input of the single attention layer is a set of node features, $\boldsymbol{h} = \{\overrightarrow{h}_1, \overrightarrow{h}_2, ..., \overrightarrow{h}_N\}$, $h_i \in \mathbb{R}^F$, where $N$ is the number of nodes, and $F$ is the number of features in each node. The GAT layer updates a new set of node features, $\boldsymbol{h}' = \{\overrightarrow{h}'_1, \overrightarrow{h}'_2, ..., \overrightarrow{h}'_N\}$, $h'_i \in \mathbb{R}^{F'}$. Then a learnable linear transformation and self-attention is performed on the nodes (a shared attention mechanism $a : R^{F'} \times R^{F'} \rightarrow R^F$ computes attention coefficients):

$$e_{ij} = a(\boldsymbol{W}\overrightarrow{h}_i, \boldsymbol{W}\overrightarrow{h}_j), \tag{3}$$

which represents the importance of node $j$ to node $i$. Afterwards, the attention coefficients are normalized by the softmax function:

$$\alpha_{ij} = \text{softmax}_j(e_{ij}) = \frac{\exp(e_{ij})}{\sum_{k \in N} \exp(e_{ij})}, \tag{4}$$

Following (Veličković et al., 2018), the attention mechanism $a$ is a single-layer feedforward neural network, parametrized by weight matrix $\overrightarrow{\boldsymbol{a}} \in \mathbb{R}^{2F'}$. After applying the LeakyReLU nonlinearity, the coefficients are also expressed as:

$$\alpha_{ij} = \frac{\exp(\text{LeakyReLU}(\overrightarrow{a}^T[\boldsymbol{W}\overrightarrow{h}_i \| \boldsymbol{W}\overrightarrow{h}_j]))}{\sum_{k \in N_i} \exp(\text{LeakyReLU}(\overrightarrow{a}^T[\boldsymbol{W}\overrightarrow{h}_i \| \boldsymbol{W}\overrightarrow{h}_k]))}, \tag{5}$$

where $(\cdot)^T$ represents the transposition operations and $\|$ means the concatenation. Then the final output of each node is updated on the strength of the similar neighborhood LR feature nodes $\overrightarrow{h}_j$:

$$\overrightarrow{h}'_i = \sigma\left(\sum_{j \in N} \alpha_{ij}\boldsymbol{W}\overrightarrow{h}_j\right), \tag{6}$$

We also employ the content branch to recover texture details shown in Fig. 1, which is a stack of 3 deconvlutional and 3 convolutional layers.

### 2.3. Feedback Mechanism

The feedback mechanism is a loop iteration to allow the network to correct previous states and regenerate high-level representations. Such iterative cause-and-effect process helps to achieve the principle of the feedback scheme for image SR: high-level information can guide an LR image to recover a better SR image (Li et al., 2019). In our network, we utilize the feedback mechanism to transfer the feature summation with high-level information got from two branches to the low-level information of an input $x$. The judgment of the feedback connection controller (shown in Fig. 1) determines the time ($T$) of the feedback iteration, also named the feedback connection. The feedback mechanism is the recurrent CNN structure to refine $x$ features with high-level $x^T$ by the feedback connections ($T - th$ iteration). It can be unfolded to $T$ iteration, in which each iteration $t$ is temporally ordered from 1 to T. The hidden state of each iteration is tied with the loss function and the weight parameters of each iteration are shared. The input of t-th iteration receives the feedback information t-1 iteration to correct original low-level inputs.

Table 1: Quantitative results of cross-protocol super-resolution and artifacts removal tasks. The best results are highlighted in bold.

| Methods | Super-Resolution | | Artifacts Removal | |
|---|---|---|---|---|
| | PSNR | SSIM | PSNR | SSIM |
| Bicubic | 27.34±1.32 | 0.8882±0.0232 | 22.58±3.59 | 0.6855±0.1345 |
| SRCNN (Dong et al., 2014) | 29.46±1.68 | 0.9042±0.0796 | 24.68±3.38 | 0.7294±0.1216 |
| VDSR (Kim et al., 2016) | 29.66±1.18 | 0.9026±0.0731 | 25.39±2.72 | 0.7588±0.0921 |
| EDSR (Lim et al., 2017) | 30.23±1.56 | 0.9145±0.0229 | 25.68±3.61 | 0.7824±0.0952 |
| DDBPN (Haris et al., 2018) | 30.34±1.56 | 0.9171±0.0208 | 25.58±3.56 | 0.7821±0.0952 |
| FB-GACN (Ours) | **30.48**±1.63 | **0.9185**±0.0194 | **25.78**±3.71 | **0.7839**±0.1003 |

## 3. Experimental Results

### 3.1. Datasets

Two experiments were conducted to evaluate the performance of the feedback graph attention convolutional network. The first experiment is solving a cross-protocol super-resolution problem on diffusion MRI data (MUSHAC) (Tax et al., 2019). The HR images were obtained by state-of-the-art diffusion MRI acquisition by Prisma scanner with voxel size (1.5 $\times$ 1.5 $\times$ 1.5 $mm^3$), and the corresponding LR images were scanned by the standard acquisition of Prisma with a larger voxel size (2.4 $\times$ 2.4 $\times$ 2.4 $mm^3$). Nine subjects are used as training set and one subject for testing. For the second experiment, we utilize the proposed network to remove the MRI artifacts and regenerate HR images by the scale $\times 2$. We randomly divided the public WMH dataset (Kuijf et al., 2019) into training (2225 images from 48 patients), validation (278 images from 6 patients) and test parts (278 images from 6 patients). Afterward, the simulated artifacts of FLAIR modality (Kuijf et al., 2019) were generated by the physical model of MRI motion artifacts.

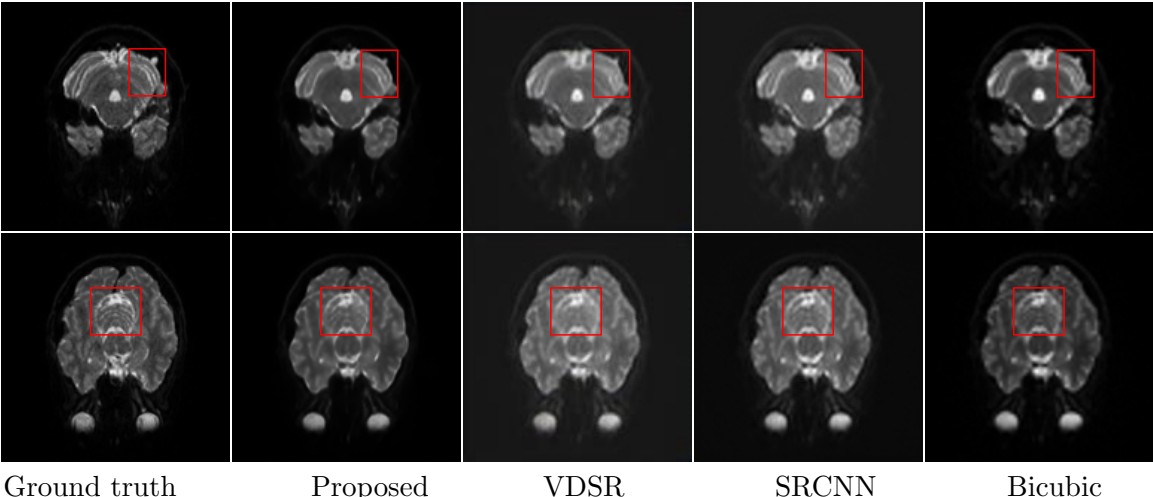

| Ground truth | Proposed | VDSR | SRCNN | Bicubic |

Figure 3: Comparison with state-of-the-art methods of cross-protocol super-resolution on the diffusion MRI data (MUSHAC). Best viewed by zooming in on the screen.

### 3.2. Implementation Details

In each training batch, nine LR patches are randomly extracted as inputs. We train our model 300 epochs with ADAM optimize and learning rate is set as $10^{-4}$ initially and is divided by 2 every 80 epochs. We implement experiments with PyTorch using a NVIDIA TITAN X GPU.

### 3.3. Comparisons with State-of-the-Art Methods

In order to evaluate the performances of our algorithms, we compare them with the start-of-the-art methods qualitatively and quantitatively. The four most recent state-of-the-art super-resolution methods are listed as follows: the Very Deep Super Resolution Network (VDSR) from (Kim et al., 2016), the Super-Resolution Convolutional Neural Network (SR-CNN) from (Dong et al., 2014),the Enhanced Deep Residual Networks (EDSR) from (Lim et al., 2017), and the Deep Back-Projection Networks For Super-Resolution (DBPN) from (Haris et al., 2018). We use open-resource implementations from the authors and train all the networks on the same dataset for a fair comparison.

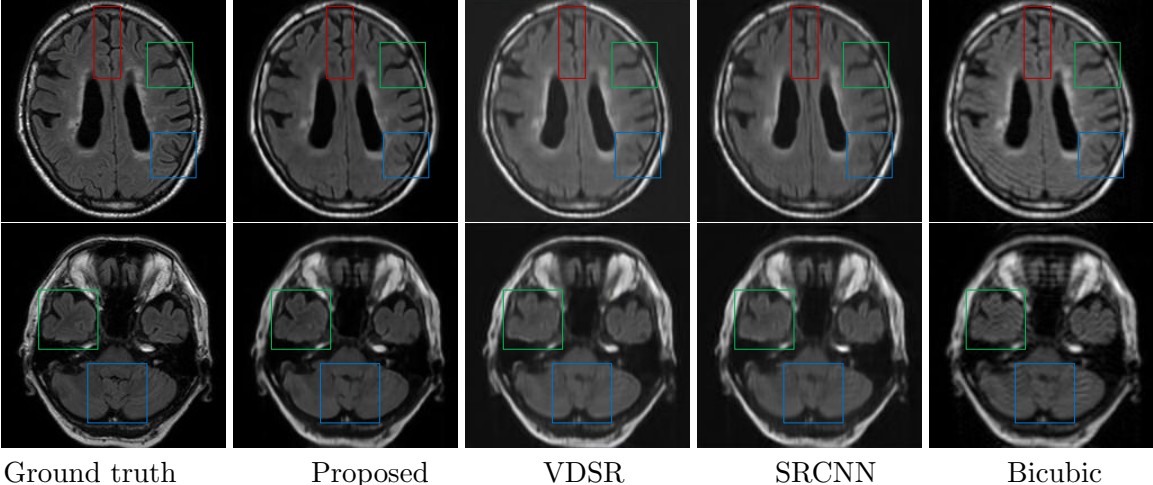

| Ground truth | Proposed | VDSR | SRCNN | Bicubic |

Figure 4: Comparison with state-of-the-art methods of artifacts removal with magnification factors ×2 and the input size 100×100. Best viewed by zooming in on the screen.

### 3.4. Quantitative Results

The quantitative evaluation of the network using the peak signal-to-noise ratio (PSNR) and the structural similarity (SSIM) scores are listed in Table 1.

**Cross-Protocol Super-Resolution**: This task is to evaluate the the performance of our method on the cross-protocol diffusion MRI quality enhancement. Our method achieves better results in comparison with other state-of-the-art methods, especially 3.46 dB higher than the traditional bicubic interpolation method.

**Artifacts Removal**: To verify the effectiveness of our proposed network towards removing MRI artifacts and super-resolution scale ×2, the PSNR and SSIM results of MRI artifacts are listed in Table 1. Our method outperforms all the state-of-the-art algorithms with the best PSNR 25.78 dB and SSIM 0.7839.

### 3.5. Qualitative Evaluation

**Cross-Protocol Super-Resolution**: The qualitative results of our methods on the diffusion MRI data (MUSHAC) by the standard and the start-of-the-art acquisition of Prisma are shown in Figure 6. It can be observed that our proposed method obtains higher visual quality and recovers clearer structures with finer contrast.

**Artifacts Removal**: The qualitative results of our methods at magnifications ×2 with artifacts are shown in Figure 5. It can be observed that our proposed method can remove artifacts and obtain the super-resolution results from the LR images. It recovers clearer structures with finer contrast, edges and lesion information.

### 3.6. Ablation study

Table 2: Ablation study results (PSNR/SSIM): Comparisons our proposed model with the configuration without (w/o) the graph similarity knowledge.

| Ablation configuration | Super-Resolution | Artifacts Removal |
|---|---|---|
| w/o graph similarity | 30.35/0.9177 | 25.65/0.7735 |
| ours | 30.48/0.9185 | 25.77/0.7835 |

**Graph similarity knowledge**: We conduct an ablation study to demonstrate the effectiveness of the graph similarity branch. We compare the proposed network with and without patch-based similarity knowledge in terms of PSNR and SSIM on the test data, shown in Table 2. The graph similarity branch boosts the performance both in the super-resolution and artifacts removal tasks.

**Feedback Mechanism**: We explore the effect of the iterative number of feedback connections. It can be observed from Table 3 that the reconstruction performance is improved when the iterative number increases from $T = 1$ to $T = 4$. Considering the balance between the computational time and the performance, $T = 4$ is chosen as the iterative number in our paper.

Table 3: The impact of the iterative number $T$ of feedback connection.

| Feedback Connection | T=1 | T=2 | T=3 | T=4 |
|---|---|---|---|---|
| Super-Resolution | 30.22/0.9172 | 30.28/0.9173 | 30.34/0.9177 | 30.48/0.9185 |
| Artifacts Removal | 25.26/0.7632 | 25.41/0.7647 | 25.49/0.7682 | 25.77/0.7835 |

### 4. Conclusion

In this paper, we proposed a novel feedback graph attention convolutional network to enhance the visual quality and remove the common distortions (e.g., artifacts) of MR images, considering the self-similarity and correlations across MRI sub-regions. We regard each sub-region as a node and construct a graph to capture the global structure. We employ the feedback mechanism to recover texture details by refining low-level representations with high-level information in a time-series way. Comprehensive qualitative and quantitative experiments show that our algorithm can remove artifacts and further generate high-resolution MRI with finer structure, contrast and lesion information.

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

## Appendix A. Brats experiments

Considering the patients of MUSHAC datasets are limited, we also verified our models on BRATS 2018 public dataset with noise and blur for super-resolution problem. We generated the LR images with the same method used in [1, 2] while the original MRIs were used as the HR ground truth images. 200 patients, 25 patients, and 25 patients are used as the training, validation, and test data, respectively. For each patient, we picked 25 most informative slices. Our model also achieves the best performance (34.486/ 0.956) than other baselines (e.g.,EDSR (33.989 /0.953), DDBPN(34.264/0.954), VDSR(32.742/0.945), SRCNN(27.195/0.918), Bicubic (22.818 0.882)).

## Appendix B. Training details

In each training stage, nine LR patches with the size of 50x50 are randomly extracted as inputs. The number of neighboring nodes is set as 5. We train our model 300 epochs with ADAM optimize and learning rate is set as 104 initially and is divided by 2 every 80 epochs. We implement experiments with PyTorch using a NVIDIA TITAN X GPU. The total training process takes almost 18 hours and can process a 256x256 image within 0.2s.

## Appendix C. Qualitative Evaluation

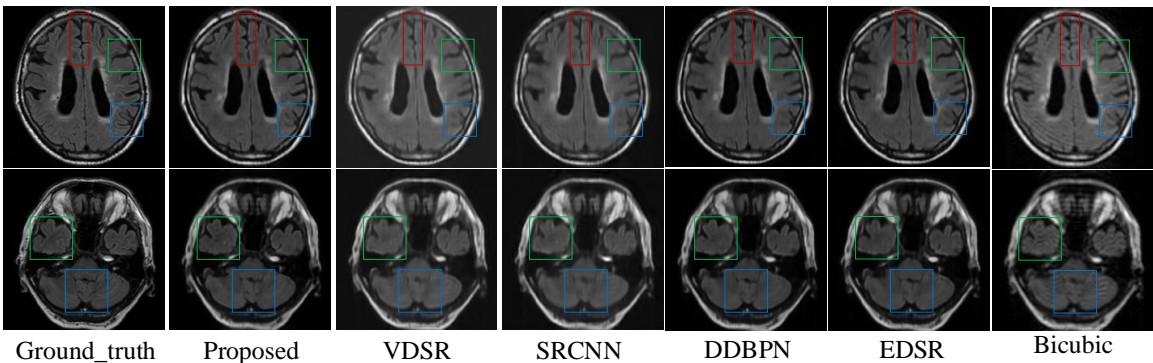

| Ground_truth | Proposed | VDSR | SRCNN | DDBPN | EDSR | Bicubic |

Figure 5: Comparison with state-of-the-art methods of artifacts removal with magnification factors ×2 and the input size 100×100. Best viewed by zooming in on the screen.

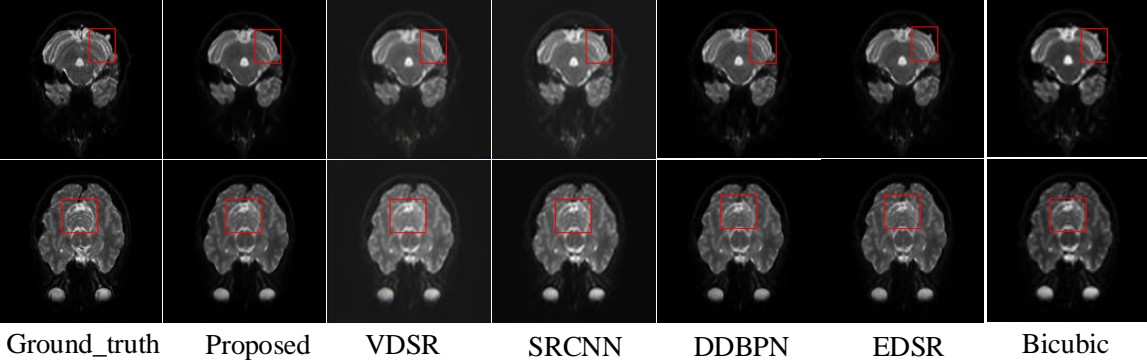

| Ground_truth | Proposed | VDSR | SRCNN | DDBPN | EDSR | Bicubic |

Figure 6: Comparison with state-of-the-art methods of cross-protocol super-resolution on the diffusion MRI data (MUSHAC). Best viewed by zooming in on the screen.

