# OpenReview forum: "Feedback Graph Attention Convolutional Network for MR Images Enhancement by Exploring Self-Similarity Features"
_MIDL.io/2021/Conference — MIDL 2021_

### Official Review · AnonReviewer4 · 2021-03-01

**Confidence:** 5
**Preliminary Rating:** 1
**Final Rating:** 3

**Summary:**

In this paper, graph attention networks (GAT) are merged with a conventional CNN to structure a new CNN architecture for MR reconstruction. A feedback mechanism which progressively refines the output is also adopted from SRFBN was also adopted. All in all, the paper is poorly written, lacking proper details of their novel contribution and also lacking the necessary comparison/ablation studies that should be performed.

**Strengths:**

Attention schemes are of high interest nowadays, especially owing to the overwhelming interest in Transformers and self-attention used in various fields. This paper is among the very few which adopted *graph attention* for MR reconstruction.

**Weaknesses:**

1.  The authors claim three main novelties of the work. These can all be summarized into one sentence - novel network architecture which utilizes graph attention and feedback loops. Although authors do try to explain why graph attention is useful, it still lacks motivation, and I do not see a clear reason why graph attention should be used in the network. Channel attention used in [1], or self attention used in [2] should suffice for euclidean MR data. Symmetry/Self-similarity features can equally be exploited using other attention schemes. What is the reason for encoding euclidean data to a graph? This should clearly be stated.

2. Methods section should provide enough details so that the readers have no difficulty reproducing the method. Although it is quite clear how the authors constructed the graph similarity branch, the *2.3 Feedback Mechanism* lacks detail on how the feedback loop was implemented. Does it use RNNs as in the original work? Does it share the parameters before/after the feedback?

3. Comparisons with state-of-the-art methods are not really state-of-the-art as the authors claim. VDSR and SRCNN are now considered outdated. I suggest that the authors compare with methods such as SRFBN since they adopted feedback loops from the work, and other CNN models which employ attention mechanisms. Moreover, Fig. 2 and 3 should be comparing the reconstructed results using EDSR and DDBPN, which they show at Table 1 that they are better than VDSR and SRCNN.

4. Ablation study should also contain a part where graph attention layers are replaced with other types of attention - spatial/channel attention, and verify its superiority.






[1] Lee, Joonhyung, et al. "Deep learning fast MRI using channel attention in magnitude domain." 2020 IEEE 17th International Symposium on Biomedical Imaging (ISBI). IEEE, 2020.

[2] Wu, Yan, et al. "Self-attention convolutional neural network for improved MR image reconstruction." Information sciences 490 (2019): 317-328.

**Deanonymize Review:**

no

**Detailed Comments:**

1. Introduction

1)

*the thick slices and low scan time MRI images have to be utilized to get a desired signal-to-noise ratio*.

This is a rather bold statement, and needs rephrasing.

2)

While describing the current art in using DL for super-resolution, [1] is introduced, when it is not a deep learning method.

3)

*Nowadays, it is an interesting trend to combine GNN and CNN to develop their corresponding advantages*

Give the readers at least 2-3 references which show that there are actual interests in combining CNN/GNN together. The reference is currently pointing to GAT.

4)

* For most conventional SR algorithms, ~ *

The sentence is baffling. I would suggest rephrasing.


2. Method

1) I do not see which part of the net architecture is **specialized** for MR images. Please elaborate.



[1] Meurée, Cédric, et al. "Patch-based super-resolution of arterial spin labeling magnetic resonance images." Neuroimage 189 (2019): 85-94.

**Final Rating Justification:**

I appreciate the authors' efforts to fully address my comments and concerns. Although some of my comments encouraged the authors to perform a comparison/ablation study, I understand that extensive experiments are hard to perform given the amount of time given for rebuttal. Other than that, since most of my concerns were about the motivation of the work, It is much clearer why graph attention could be more beneficial than using channel attention or spatial attention by being able to exploit self-similarity features. It is now easier to understand how the feedback mechanism was implemented, and now I believe that the work is of good quality to contribute to the community.

**Justification Of The Preliminary Rating:**

Motivation for the network design is weak. Rigorous comparisons/ablation studies need to be added. Proper details about the implementation should be included. If all the above questions and problems are addressed, especially the concern about the motivation for using GAT, I think the paper has the potential for contributing in the MR recon. community.

**Paper Type:**

validation/application paper

**Questions To Address In The Rebuttal:**

1. Motivation for using graph attention should be strengthened.

2. Details about the implementation of feedback mechanism should be given.

**Special Issue:**

no

---

> ### Author Response · Authors · 2021-03-16
> **Response to reviewer 4**
>
> We thank the reviewer for the careful reading of our manuscript and the constructive comments. We address all questions in detail below.
>
> Q1: ‘Channel attention used in [1], or self attention used in [2] should suffice for euclidean MR data. Symmetry/Self-similarity features can equally be exploited using other attention schemes. What is the reason for encoding euclidean data to a graph? This should clearly be stated. What is the reason for encoding euclidean data to a graph? This should clearly be stated.’
>
> Essentially, different from other channel/spatial attention to give more focuses on the informative components and suppress the non-informative part, we regarded each patch as a node and updated the features by calculating the similarity between patch-based node features. Attention mechanism deals with the channel-based or spatial-based features while ours’ graph is based on the patch-based features. Though channel/spatial attention is a powerful tool, it is not capable of calculating the similarity among patch-based features and then merging high-similarity information.
>
> Q2: ‘Motivations and specialized design for MRI’
>
> Our method aims to learn the symmetry and self-similarity relationship of  patch-based features in multi-modal brain MR images where the structure of the brain is normally symmetry. To meet this requirement, we designed a specialized Graph-based structure to merge the high-similarity information of sub-regions by updating larger weights to the more important and similar nodes or features in a graph attention fashion.  We have added the claim into the Page 3 and updated the manuscript.
>
> Q3: Details of feedback mechanism: the 2.3 Feedback Mechanism lacks detail on how the feedback loop was implemented. Does it use RNNs as in the original work? Does it share the parameters before/after the feedback?
>
> The feedback mechanism is the recurrent CNN structure to refine x features with high-level xT by the feedback connections (T-th iteration). It can be unfolded to T iteration, in which each iteration t is temporally ordered from 1 to T.  The hidden state of each iteration is tied with the loss function and the weight parameters of each iteration are shared. The input of t-th iteration receives the feedback information t-1 iteration to correct original low-level inputs.  We have added it into 2.3 subsection and updated the manuscript.
> More details and information of feedback mechanism can refer to the following work:
>
> [1] Feedback Network for Image Super-Resolution
>
> Q4: Interests in combination of CCN and GNN:
>
> GNNs help with reducing the data dimensionality from image features extracted by CNN to high-level and compact features in graph nodes.  FCNs are limited in the receptive field. Adding a GNNs could increase the receptive field of networks when dealing with large images.
> The combination of CCN and GNN is a convolutional graph neural network that generalizes the operation of convolution from grid data to graph data. Since ConvGNNs stack multiple graph convolutional layers to extract high-level node representations, it plays a central role in building up many other complex GNN models [2].  We have added it into 2.2 subsection and updated the manuscript.
>
> [2]. A Comprehensive Survey on Graph Neural Networks
>
> Q5: Sentence rephrasing and providing visual results of EDSR and DDBPN
>
> Thanks for your comments. We have already rephrased all mentioned sentences and added more visual results into our appendix.

---

### Official Review · AnonReviewer2 · 2021-03-08

**Confidence:** 4
**Preliminary Rating:** 3
**Recommendation:** Poster

**Summary:**

The author propose to include a graph network to consider the global structure for MR Images Enhancement. They also propose a feedback mechanism with a recurrent structure to refine low-level representations to obtain more texture details. Specifically, they use self-similarity learning strategy to update the features of each node in a graph. The work is highly related to the work graph attention networks(GAT)[Velickovic et al.,2018], but adapted to the problem of MR Images Enhancement. The results show an improved performance of the method over state-of-the-art and an ablation study was performed to justify the model’s design.

**Strengths:**

It was well written, and the proposed method is sound. The idea of graph attention convolutional network and feedback mechanism is actually not new. However, it is good to use this idea for medical image enhancement, and the experiments in the paper showed some improvement of the method.

**Weaknesses:**

One limitation is the small dataset (10 subjects in total, 9 used for training) for the first experiment. It should be tested on more data and also different datasets. Information about the training process is missing. How difficult is the model to train?

**Deanonymize Review:**

no

**Detailed Comments:**

The figure is not clear and hard to interpretate.  They should be improved.

**Justification Of The Preliminary Rating:**

It is interesting to apply the graph attention convolutional network and feedback mechanism, to improve MRI quality, which contributes to acceptance of the manuscript. However, the validation work of this paper is not complete, and they should be better performed. Based on this two points, we recommend a weak accept.

**Paper Type:**

validation/application paper

**Questions To Address In The Rebuttal:**

It should be tested on more data and also different datasets. Information about the training process is missing.

**Special Issue:**

no

---

> ### Author Response · Authors · 2021-03-16
> **Response to reviewer 2**
>
> We thank the reviewer for valuable comments.
>
> Q1: One limitation is the small dataset (10 subjects in total, 9 used for training) for the first experiment. It should be tested on more data and also different datasets.
>
> Considering the patients of MUSHAC datasets are limited, we also verified our models on BRATS 2018 public dataset with noise and blur for super-resolution problem. We generated the LR images with the same method used in [1, 2] while the original MRIs were used as the HR ground truth images. 200 patients, 25 patients, and 25 patients are used as the training, validation, and test data, respectively. For each patient, we picked 25 most informative slices. Our model also achieves the best performance (34.486/ 0.956) than other baselines (e.g.,EDSR(33.989/0.953), DDBPN(34.264/0.954), VDSR(32.742/0.945), SRCNN(27.195/0.918), Bicubic(22.818 0.882)). We have added this experiment into our appendix.
>
> [1] LRTV: MR image superresolution with low-rank and total variation regularizations
>
> [2] MR Image Super-Resolution via Wide Residual Networks with Fixed Skip Connection
>
> Q2: Information about the training process is missing. How difficult is the model to train?
>
> In each training stage, nine LR patches with the size of 50x50 are randomly extracted as inputs.  The number of neighboring nodes is set as 5. We train our model 300 epochs with ADAM optimize and learning rate is set as 10−4 initially and is divided by 2 every 80 epochs. We implement experiments with PyTorch using a NVIDIA TITAN X GPU. The total training process takes almost 18 hours and can process a 256x256 image within 0.2s. We have added this explanation into our appendix.
>
> Q3: The figure is not clear and hard to interpretate. They should be improved.
>
> We will improve the image resolution in our final manuscript.

---

### Official Review · ~Simeon_Emilov_Spasov1 · 2021-03-08

**Confidence:** 4
**Preliminary Rating:** 3
**Recommendation:** Oral
**Final Rating:** 4

**Summary:**

The goal of the paper is to create a deep learning method for improving medical image quality by exploiting global structural information and self-similarity between different image patches. The method consists of three main parts: 1) two parallel processing branches to enhance texture detail. One comprises conventional convolutions and the other uses graph convolutions to integrate information from different image sub-regions based on feature similarity; 2) A recurrent feedback mechanism to iteratively refine the reconstruction quality; 3) reconstruction based on upsampling blocks. The authors provide favourable results against SOTA image reconstruction methods.

**Strengths:**

1.	The paper is technically novel with strong motivation. Exploiting image patch similarity using graph attention convolutions is interesting and is demonstrated to have advantages.
2.	Write-up is in general clear (text and figures) other than section 2.2 Graph Similarity Branch.
3.	Ablation studies are necessary and good. They show the advantage of having the graph convolutions exploiting the similarity between image patches and integrating more global information in patch reconstruction as well as the recurrent feedback mechanism.
4.	Results against existing SOTA methods are strong.


**Weaknesses:**

1.	One issue is that the authors need to add standard deviations to tables 1,2,3. Did the authors repeat their experiments? It is *key* to show statistical significance of results.
2.	I suggest the authors modify section 2.2 Graph Similarity Branch and say that they form a graph using the n x d matrix where they assume there are n nodes with d-dimensional features. Then, do the authors assume this graph is fully connected? The connectivity of this graph is never explicitly stated. Do the authors calculate attention coefficients between each pair of i and j nodes? Further, after the GAT layers, I assume the authors reshape back to h/s x w/s x d and continue with deconvolution as implied by figure 1? I suggest adding this explanation in full to the section.
3.	Need to define the acronyms LR (low resolution) and HR (high resolution) before the authors use them.


**Deanonymize Review:**

yes

**Detailed Comments:**

Could you please comment on the statistical significance of your results (standard deviations in tables 1,2,3)?

**Final Rating Justification:**

I thank the authors for incorporating my suggestions for 1) adding standard deviations of results and 2) explaining in greater detail how a graph is formed based on neighbours in feature space. I believe using graph representation learning to transfer global information between image patches would be interesting to the MIDL community despite the overlap of PSNR and SSIM results given the newly added stds.

**Justification Of The Preliminary Rating:**

The paper is technically novel in that it practically transfers more global image information to local image patches by leveraging advances in graph representation learning.  The write-up is very clear and results convincing. I suggest the authors 1) address the issues surrounding clarity of section 2.2 Graph Similarity Branch and 2) add statistical significance analysis of results.

**Paper Type:**

methodological development

**Special Issue:**

yes

---

> ### Author Response · Authors · 2021-03-16
> **Response to reviewer 3**
>
> We thank the reviewer for the positive comments.
>
> Q1: One issue is that the authors need to add standard deviations to tables 1,2,3. It is key to show statistical significance of results.
>
> We calculated the standard deviation of PSNR and SSIM listed as follows: ours (30.48±1.63, 0.9185±0.0194, 25.78±3.71, 0.7839±0.1003), bicubic(27.34±1.32, 0.8882±0.0232, 22.58±3.59, 0.6855±0.1345), EDSR (30.23±1.56, 0.9145±0.0229, 25.68±3.61 , 0.7824±0.0952),DDBPN (30.34±1.56   0.9171±0.0208   25.58±3.56   0.7821±0.0952). The first two PSNR and SSIM were calculated for super-resolution tasks and the last two were for artifacts removal task. We have already modified them in tab1 .
>
> Q2: Reorganized of Graph Similarity Branch paragraph.
>
> We will rephrase this paragraph according to your suggestion and define the acronyms LR and HR before we use them. ‘We form a graph using the n x d matrix where we assume there exist n nodes with d-th dimensional features. Each node means h/s x w/s features. Each node is connected with five neighboring nodes and the attention coefficient of each node is updated. Finally, we reshape the features back to h/s x w/s x d and process with deconvolution layer.’ We have already rephrased them in our 2.2 subsection.

---

### Official Review · AnonReviewer1 · 2021-03-09

**Confidence:** 5
**Preliminary Rating:** 3
**Recommendation:** Poster
**Final Rating:** 3

**Summary:**

This paper proposed a Feedback Graph Attention Convolutional Network (FB-GACN) that can be applied for cross-protocol super-resolution and MRI artifacts removal. In particular, they proposed a graph similarity branch along with the content branch to explore the globally structural similarity among similar paired sub-regions. Furthermore, they employed a feedback mechanism (consecutive iterations) to facilitate the final image quality.+ The paper is well written and easy to follow.
+ The proposed framework is technically sound.
+ The experiments are comprehensive.


**Strengths:**

+ The paper is well written and easy to follow.
+ The proposed framework is technically sound.
+ The experiments are comprehensive, ablation study is conducted to demonstrate the effectiveness of graph similarity branch and feedback mechanism.


**Weaknesses:**

- More citations related to MR super-resolution are needed.
- There is no visualization comparison with state-of-the-art methods. It’s better to show the EDSR and DDBPN results in figure 3, as the SRCNN and VDSR are not recently published methods.
- Compared to EDSR and DDBPN as shown in Table 1, the proposed method is not significantly better in both terms of PSNR and SSIM.


**Deanonymize Review:**

no

**Detailed Comments:**

1. In cross-protocol super resolution experiment, the authors used only one subject for testing, it’s better to use leave-one-out strategy.
2. The authors mentioned that ‘The HR images were obtained by state-of-the-art diffusion MRI acquisition by Prisma scanner with voxel size….. with a larger voxel size (2.4*2.4*2.4)”, what preprocessing steps were applied, e.g., registration and normalization. What’s relationship between LR image and bicubic image as shown in figure 3.
3. What does this mean in Section 3.2, ‘In each training batch, nine LR patches are randomly extracted as inputs.’?
4. What’s the inference time for the proposed methods compared to other methods, how does it change when more feedback iteration is applied.


**Final Rating Justification:**

I appreciate the feedback from the authors. They have addressed all the concerns listed in the comments. I would like to see this work published and code released in the future that will benefit the biomedical imaging community.

**Justification Of The Preliminary Rating:**

This paper proposed a novel feedback graph attention convolutional network to improve the image quality from low-resolution MR images or MR images with motion artifacts. It is well written and technically sound. However, the experiments to evaluate the super-resolution result might be inappropriate and the comparisons are a little insufficient.

**Paper Type:**

methodological development

**Special Issue:**

no

---

> ### Author Response · Authors · 2021-03-16
> **Response to reviewer 1**
>
> Thank you for your positive and valuable feedback.
>
> Q1: In cross-protocol super resolution experiment, the authors used only one subject for testing, it’s better to use leave-one-out strategy.
>
> In the cross-protocol super resolution experiment, limited by small-scale MUSHAC datasets, we also test our models on BRATS 2018 public dataset with noise and blur for super-resolution problem. Specifically, following [[1, 2]], LR images were generated while the original MRIs were used as the HR ground truth images. 200 patients, 25 patients, and 25 patients are used as the training, validation, and test data, respectively. For each patient, we selected 25 most informative slices. Our model also achieves the best performance (34.486/ 0.956) than other baselines (e.g., EDSR(33.989/0.953), DDBPN(34.264/0.954), VDSR(32.742/0.945), SRCNN(27.195/0.918), Bicubic(22.818 0.882)). We have added this experiment into our appendix.
>
>  [1] LRTV: MR image superresolution with low-rank and total variation regularizations
>
> [2] MR Image Super-Resolution via Wide Residual Networks with Fixed Skip Connection
>
> Q2: what preprocessing steps were applied, e.g., registration and normalization. What’s relationship between LR image and bicubic image as shown in figure 3.
>
> We down-scale the low-resolution images obtained by the standard scanner to ensure x2 scale super-resolution between low- and high-resolution size. Besides, for minimizing the variation of pixel intensities, we employed the Gaussian normalization on the pixel-intensity level. The relationship between LR images and bicubic images is x2 scale bicubic interpolation.
>
> Q3: What does this mean in Section 3.2, ‘In each training batch, nine LR patches are randomly extracted as inputs.’?
>
> The LR patches are randomly sampled from complete LR images and then are imported into the models with the batch size of nine for one iteration.
>
> Q4: What’s the inference time for the proposed methods compared to other methods, how does it change when more feedback iteration is applied.
>
> Since we share the weight of feedback block, it largely decreases the training parameters. Our network can process a 256x256 image within 0.2s. The inference time increases to 0.33s when double feedback iteration is applied.
>
> Q5: More citations related to MR super-resolution are needed.
>
> We have cited more relevant papers below.
> [3]. Lee, Joonhyung, et al. "Deep learning fast MRI using channel attention in magnitude domain." 2020 IEEE 17th International Symposium on Biomedical Imaging (ISBI). IEEE, 2020.
> [4] Wu, Yan, et al. "Self-attention convolutional neural network for improved MR image reconstruction." Information sciences 490 (2019): 317-328.
> [5] Meurée, Cédric, et al. "Patch-based super-resolution of arterial spin labeling magnetic resonance images." Neuroimage 189 (2019): 85-94.
>
> Q6: There is no visualization comparison with state-of-the-art methods. It’s better to show the EDSR and DDBPN results in figure 3.
>
> We have added visualization comparisons of EDSR and DDBPN in our appendix.

---

### Meta-Review · Area_Chair1 · 2021-03-29

**Recommendation:** Accept (Poster)

**Metareview:**

Suggest acceptance with the following revision advices
1. As the reviewers suggested, there are relevant works that need proper survey and citations, such as the works using superresolution (eg. A Comparative Study of CNN-based Super-resolution Methods in MRI Reconstruction and Its Beyond, Signal processing: image communication, Volume 81, February 2020) and image enhancement for detail preservation (eg.Iterative feature refinement for accurate undersampled MR image reconstruction", Physics in Medicine and Biology, vol. 61, p. 3291－3316, 2016), etc
2. The super resolution methods are more often not for artifact or noise removal. The authors need to properly differentiate image superresolution from artifact removal methods.

**Paper Type:**

methodological development

---

### Decision · Program_Chairs · 2021-03-31

Accept